# ZNF16 is a nucleolar-associated protein that regulates expression of rDNA and cancer-associated genes

Chelsea L. George[1], Laura A. Espinoza Quevedo[2], Jason Paratore[2], Matthew J. Alcaraz[2], Arlene P. Levario[1], Yasir Rahmatallah[3], Galina V. Glazko[3], Nathan S. Reyna[4] and Laura A. Diaz-Martinez[1,2,*]

## ABSTRACT

ZNF16 (also known as HZF1 and KOX9) is a multi-C2H2 zinc finger protein first identified via its expression in human T-cells and shown to have a role in blood cell differentiation. ZNF16 was later shown to be ubiquitously expressed in a variety of fetal and adult tissues, suggesting a broader function. In this study, we confirm the ubiquitous expression of ZNF16 in a variety of cancer and non-cancer cell lines and show that ZNF16 depletion reduces cell viability in all cell lines tested. Furthermore, we show that ZNF16 localizes to the nucleolus in a transcription-dependent manner, interacts with the intergenic spacer region of the rDNA and promotes rDNA transcription. Additionally, RNA-sequencing experiments after ZNF16 depletion revealed that ZNF16 also has roles in a variety of pathways including extracellular matrix-receptor interaction, focal adhesions, cytokine-cytokine receptor interactions, human papillomavirus infection and cancer pathways. These findings are consistent with broader roles for ZNF16, including the regulation of nucleolar function, a process that is essential for all cells, and provide evidence at the cellular/molecular level of its role in the regulation of cancer-associated genes (e.g. NRAS, BIRC3, EGFR).

KEY WORDS: Zinc finger proteins, Nucleolar function, RDNA transcription, RRNA expression

## INTRODUCTION

Zinc finger proteins represent 5-10% of the human proteome (Andreini et al., 2006; Vilas et al., 2018), with ~700 of these proteins belonging to the C2H2 zinc finger family (Schmitges et al., 2016), which comprises almost 50% of all putative transcription factors in vertebrates. These proteins typically consist of tandem arrays of zinc finger motifs that provide DNA-binding specificity and effector motifs (e.g. KRAB, SCAN) that mediate interaction with transcriptional activators or inhibitors (Schmitges et al., 2016). Although the most studied role of zinc finger motifs is in DNA binding (Klug and Rhodes, 1987), zinc finger motifs can also interact with RNA (Laity et al., 2001), proteins (Gamsjaeger et al., 2007) and

lipids (Laity et al., 2001), indicating a wide diversity of functions for these types of proteins.

ZNF16 (also called HZF1) is a C2H2 zinc finger protein that has been associated with cancer progression (Ahn et al., 2020; Lee et al., 2021; Zhang et al., 2020). ZNF16 expression was positively associated with histological grade, shorter survival, and increased risk of relapse in gallbladder carcinoma patients (Ahn et al., 2020). Genetic changes in ZNF16 have also been associated with tongue squamous cell carcinoma (TSCC), with 35% of TSCC samples having ZNF16 amplification and 5% missense mutations (Zhang et al., 2020). Consistent with a role for ZNF16 in TSCC, ZNF16 depletion reduces TSCC cell viability and xenograft tumor size (Zhang et al., 2020).

ZNF16 has 15 C2H2 zinc finger motifs in the C-terminus (Deng et al., 2010; Schmitges et al., 2016) and is localized in a zinc finger gene cluster on human chromosome 8q24.3 that contains seven zinc finger proteins (Lorenz et al., 2010). Interestingly, ZNF16 is the only protein in the cluster that lacks a Krüppel-associated box (KRAB) domain (Lorenz et al., 2010), a domain commonly associated with transcriptional repressors (Urrutia, 2003). Consistent with a role in transcription, ZNF16 is localized in the nucleus, activates transcription of a GAL1-lacZ reporter in yeast, and contains a transactivation domain in its unstructured N-terminal region that can activate transcription in yeast when fused to the GAL4-DNA binding domain (Deng et al., 2010).

ZNF16 was first identified via its expression in human T-cells (Thiesen, 1990). It was later shown to have a role in *in vitro* erythroid and megakaryocytic differentiation *in vitro* (Peng et al., 2006), and its overexpression increases cell proliferation and moderately reduces apoptosis induced by sodium arsenate (Li et al., 2011). However, the molecular mechanisms behind these functions are unclear. ZNF16 interacts with the CDK1-inhibitor INCA (Li et al., 2011) and binds upstream of the c-KIT promoter (Chen et al., 2014) in K562 cells, but ZNF16 overexpression has only minimal effects on cell cycle progression (Li et al., 2011).

In addition to its expression in blood cells, ZNF16 has also been shown to be expressed in a variety of human tissues including fetal brain, testis, cerebellum, and kidney (Lorenz et al., 2010), as well as adult brain, heart, skeletal muscle, liver, and bone marrow (Peng et al., 2006). ZNF16 interacts with proteins associated with a variety of functions including linker histones (Zhang et al., 2016), transcriptional regulators (e.g. HMGA1, DCAF7, TRIM28, HDAC1), cell signaling (e.g. MAPK1, PPP2A,GRB2), ubiquitination and protein degradation (e.g. TRIM27, USP11, PSMC3), DNA replication and repair (e.g. LIG3, MCM6), proteins found in nuclear bodies (e.g. Coilin, PML), and proteins found at the nucleolus and/or involved in rRNA processing (e.g. UBTF, TCOF1) (Schmitges et al., 2016).

Nucleoli are membraneless and highly dynamic organelles that contain over 800 proteins (Jarboui et al., 2011). They are organized around nucleolar organizing regions (NORs) that consist of multiple

[1]Department of Biological Sciences, The University of Texas at El Paso, El Paso TX 79968, USA. [2]Department of Biology, Gonzaga University, Spokane, WA 99258, USA. [3]Department of Biomedical Informatics, University of Arkansas for Medical Sciences, Little Rock, AR 72205, USA. [4]Department of Biology, Ouachita Baptist University, Arkadelphia, AR 71998, USA.

*Author for correspondence (diazmartinez@gonzaga.edu)

C.L.G., 0009-0004-6219-184X; L.A.D.-M., 0000-0002-7226-7068

tandem repeats of the rDNA gene, separated by intergenic spacers (IGS) (Potapova and Gerton, 2019; Trinkle-Mulcahy, 2018). Nucleoli are the site of rDNA transcription, rRNA processing, and ribosome biogenesis, as well as a cellular stress response center (González-Arzola, 2024; Hua et al., 2022; Núñez Villacís et al., 2018). Nucleoli number, size and activity are increased in hyperproliferative cells, serving as a prognostic marker for tumor malignancy (Trinkle-Mulcahy, 2018).

Given the expression of ZNF16 in multiple human tissues (Lorenz et al., 2010; Peng et al., 2006) and its association with a variety of proteins, including nucleolar proteins (Schmitges et al., 2016), we hypothesized that ZNF16 has a more universal role that is relevant for cells in all these tissues in addition to its specific role in blood cell differentiation. Here, we describe a novel role for ZNF16 at the nucleolus and its association with changes in expression of a variety of genes involved in cancer pathways.

## RESULTS

### ZNF16 is expressed and promotes cell viability in a variety of cell lines

ZNF16 has been reported to be expressed in a variety of human fetal and adult tissues (Lorenz et al., 2010; Peng et al., 2006). However, all studies on ZNF16 function at the cellular/molecular level have been performed in K652 leukemia cells (Chen et al., 2014; Li et al., 2011; Peng et al., 2006) or in cells expressing exogenous ZNF16 (Deng et al., 2010). To begin our study of ZNF16 functions, we first tested a panel of five non-blood cell lines for ZNF16 expression by quantitative PCR (qPCR) using probes against ZNF16 mRNA and GAPDH as internal control. ZNF16 was expressed in all five cell lines tested (Fig. 1A). Relative expression was normalized to HeLa Tet-On, which was the lowest expressing cell line. Remarkably, there was a wide range of ZNF16 expression, with the osteosarcoma cell line U2OS having 30-fold higher expression than HeLa. The two colon cancer cell lines (Hct116 and DLD1) and the non-cancer RPE1-hTERT cell line (RPE1) had intermediate levels of expression. These results indicate that ZNF16 is ubiquitously expressed in both cancer and non-cancer cell lines, suggesting that it has other potential functions, in addition to blood cell differentiation.

To begin addressing the role of ZNF16, three independent siRNAs were transfected into the highly expressing U2OS cell line, and their knockdown efficiency was quantified by qPCR. siZNF16-E most consistently reduced ZNF16 mRNA compared to the other two siRNAs (Fig. 1B), and the effectiveness of the knockdown was confirmed by western blotting (Fig. 1C). Thus, siZNF16-E was used for all subsequent experiments.

To test whether ZNF16 depletion affected cell viability, a sub-panel of cell lines was transfected with a non-targeting siRNA (siControl or siCtrl) or siZNF16-E siRNAs (Fig. 1D). Hct116 was included and DLD1 was omitted for simplicity, given that both are colon cancer cell lines and express similar levels of ZNF16 (Fig. 1A). Cell viability was markedly reduced in all cell lines treated with siZNF16-E compared to siControl (Fig. 1D). Note that the samples are normalized to an untreated sample, which is why the siControl samples are not all at 100% viability.

### ZNF16 is enriched at the nucleoli

To begin studying ZNF16 function, we first asked where ZNF16 is located in cells. First, we visualized ZNF16 localization by transient transfection of plasmids containing EGFP (control), EGFP-ZNF16 (single EGFP+ZNF16) or 3xEGFP-ZNF16 (triple EGFP+ZNF16) in HeLa and U2OS cells, followed by immunostaining with antibodies against the nucleolar protein Ki67 (Fig. 2A,B). Although all exogenous forms of ZNF16 localize to the nucleus, the exact localization within subnuclear structures varies depending on the tag: 3xEGFP-ZNF16 is localized to the nucleoli in both cell lines (Fig. 2A, bottom row; Fig. 2B, bottom two rows), while the localization of EGFP-ZNF16 varies by cell line. EGFP-ZNF16 localizes to the nucleoplasm in both cell lines (Fig. 2A,B, second row in both sections) and is enriched in the nucleoli in U2OS cells (Fig. 2A) but not in HeLa cells (Fig. 2B). These results indicate that the size of the N-terminal tag and/or the number of EGFP units might affect ZNF16 folding and thus its localization. Furthermore, transient

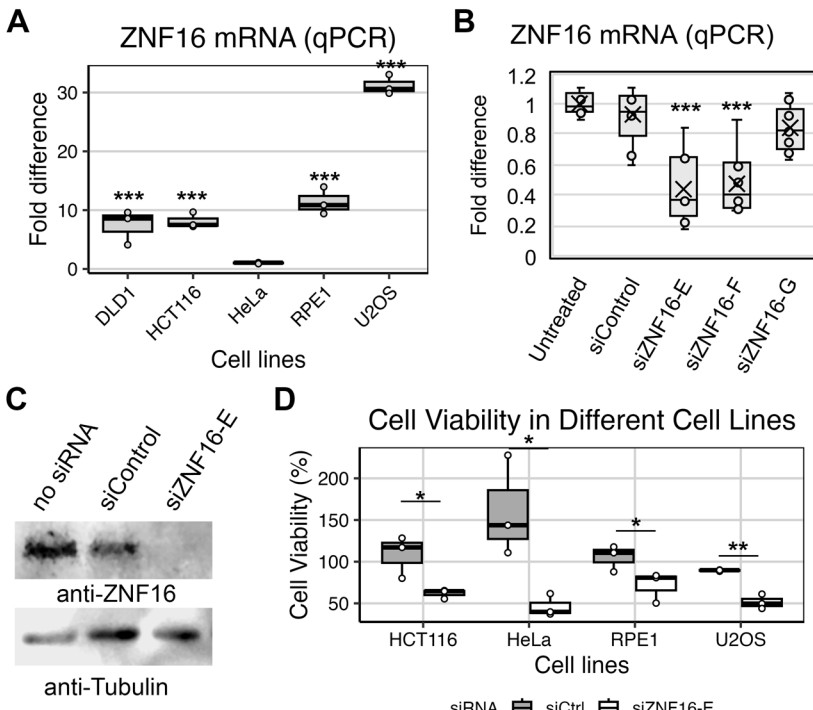

**Fig. 1. ZNF16 is expressed and promotes cell viability in different cell lines.** (A) Quantification of ZNF16 mRNA by qPCR in the indicated cell lines. Fold difference normalized to average in HeLa cells. At least three independent repeats per cell line. Each cell line was compared to HeLa by t-tests. Asterisks denote P-values as follows: *P<0.05, **P<0.01, ***P<0.001. (B) Quantification of ZNF16 mRNA by qPCR in U2OS cells treated with the indicated siRNAs for 72 h. Results from three independent experiments. (C) Corresponding representative western blot. (D) Cell viability of indicated cell lines 72 h after transfection with the indicated siRNAs. Results from each treatment were normalized to the corresponding untreated sample (100% viability, not shown). siZNF16 samples for each cell line were compared to their corresponding siCtrl sample via t-test.

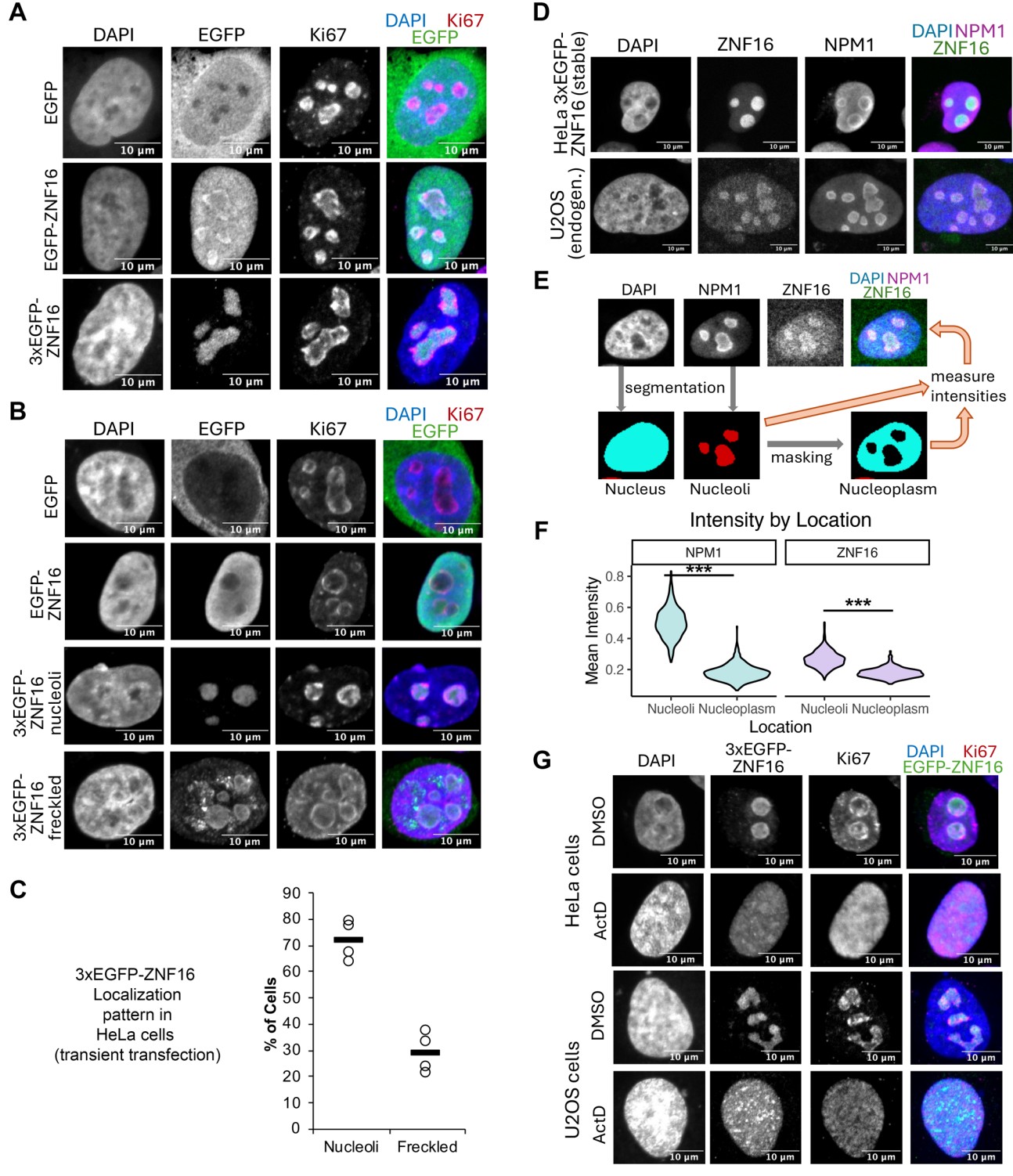

**Fig. 2. ZNF16 is enriched at the nucleoli.** (A,B) Representative micrographs of U2OS (A) and HeLa (B) cells transiently transfected with the plasmids containing EGFP, EGPF-ZNF16, or 3xEGFP-ZNF16, and immunostained with the indicated antibodies. (C) Quantification of the localization patterns after transfection of 3xEGFP-ZNF16 in HeLa cells. Four repeats of the experiment (circles) and the mean (line) shown. (D) Representative micrographs of HeLa cells stably transfected with 3xEGFP-ZNF16 (top row) and U2OS cells (bottom row) immunostained with the indicated antibodies. (E) Schematic showing the image analysis performed in CellProfiler to measure the mean intensity of ZNF16 and NPM1 signal in the nucleoli and nucleoplasm at the single cell level. Cells were segmented in the DAPI and NPM1 channels to obtain masks for the nucleus and nucleoli, respectively. The nucleoli masks were then subtracted from the nuclei masks to obtain a nucleoplasm mask. Mean intensities were then measured using the nucleoli and nucleoplasm masks per cell. (F) Violin plot showing the mean intensity signal of NPM1 and ZNF16 in the nucleoli versus nucleoplasm ($n$=507 cells). Comparisons between nucleoli and nucleoplasm localization per antibody were performed via $t$-test. Asterisks denote $P$-values as follows: ***$P$<0.001. (G) Representative micrographs of cells transfected with 3xEGFP-ZNF16 plasmid, incubated for 24 h, then incubated with 40 µM actinomycin D for 4 h before fixation and immunostaining with the indicated antibodies.

transfection of 3xEGFP-ZNF16 also resulted in two main patterns of localization: a majority of the cells (71.5% on average from four independent experiments, Fig. 2C) showed 3xEGFP-ZNF16 enrichment in the nucleoli with little localization to the nucleoplasm (Fig. 2B, third row), while the remaining cells showed both nucleolar localization and localization to other subnuclear structures (Fig. 2B, bottom row). Importantly, 3xEGFP-ZNF16 localization to the nucleoli is the dominant phenotype (Fig. 2C). These results indicate that exogenous ZNF16 localization is influenced by both the tag size and the level of expression.

In order to avoid potential artifacts due to protein overexpression, we obtained a HeLa cell line stably transfected with the 3xEGFP-ZNF16 plasmid. This cell line expresses 3xEGFP-ZNF16 at lower levels than transient transfection (data not shown). Immunostaining of this stable cell line with antibodies against the nucleolar protein nucleophosmin 1 (NPM1; Fig. 2D, top row) shows enrichment of 3xEGFP-ZNF16 at the nucleoli with little to no 'freckles' in the nucleoplasm. Lastly, immunostaining of endogenous ZNF16 in U2OS cells with anti-ZNF16 antibodies shows a similar pattern of nucleolar enrichment (Fig. 2D, bottom row). These results indicate that the pattern of 3xEGFP-ZNF16 enrichment at the nucleolus closely reflects the pattern observed for endogenous ZNF16. To quantify the intensity of ZNF16 in the nucleoli versus the nucleoplasm, U2OS cells immunostained with anti-ZNF16 and anti-NPM1 antibodies were segmented and analyzed using CellProfiler (Stirling et al., 2021). First, the nucleus was segmented in the DAPI channel, followed by segmentation of the nucleoli in the NPM1 channel. Then, the nucleoli regions were subtracted from the nucleus by generating a nucleoplasm mask. The mean intensity of NPM1 and ZNF16 was measured using the nucleoli and the nucleoplasm masks at the single cell level (see schematic in Fig. 2E). This analysis revealed that the mean ZNF16 signal is significantly higher in the nucleoli than the nucleoplasm (Fig. 2F), following a pattern similar to NPM1, a protein that is known to be localized to the nucleoli (Schmidt-Zachmann et al., 1987).

Lastly, HeLa and U2OS cells transiently expressing 3xGFP-ZNF16 were incubated with the RNApol-I inhibitor actinomycin-D to test the effect of nucleolar transcription inhibition on ZNF16 localization (Fig. 2G). After treatment with actinomycin-D, ZNF16 and the nucleolar protein Ki67 relocalize from the nucleolus to the nucleoplasm. Taken together, these results indicate that ZNF16 and exogenously expressed 3xEGFP-ZNF16 are enriched in the nucleolus, and their nucleolar localization is disrupted after inhibition of RNApol-I transcription.

## ZNF16 regulates rDNA expression

Given that both endogenous ZNF16 and 3xEGFP-ZNF16 localize to the nucleus (Fig. 2) and are enriched in the nucleoli compared to the nucleoplasm (~1.5-fold for endogenous ZNF16, Fig. 2F), we next asked whether ZNF16 regulates nucleolar function by evaluating its role in rDNA expression. First, we tested whether transient expression of exogenous ZNF16 impacts transcription of an rDNA luciferase reporter. HeLa and U2OS cells were transiently co-transfected with plasmids containing EGFP-ZNF16 or 3xEGFP-ZNF16 and the pHrD-IRES-Luc plasmid (Ghoshal et al., 2004), which contains the human rDNA promoter fused to a luciferase reporter. Expression of 3xEGFP-ZNF16, which was shown to be enriched at the nucleolus (Fig. 2A,B), significantly increased luciferase activity in both U2OS and HeLa cells (Fig. 3A,B). In contrast, EGFP-ZNF16 expression did not increase the activity of the rDNA promoter in U2OS (Fig. 3B) and increased it only slightly in HeLa cells (Fig. 3A), suggesting that the ability of 3xEGFP-ZNF16 to localize to the nucleolus might be important for its ability to activate the rDNA promoter. Furthermore,

this effect is specific to the rDNA promoter since a similar experiment using a luciferase reporter driven by a cytomegalovirus promoter (CMV) resulted in a decrease in luciferase activity after expression of 3xEGFP-ZNF16 in both cell lines (Fig. 3A,B).

The role of endogenous ZNF16 in rDNA transcription was tested via a 5-ethynyl uridine (5-EU) incorporation assay. U2OS or HeLa cells were incubated with siControl or siZNF16-E siRNAs for 72 h. Then, the cells were incubated with the nucleoside analog 5-EU, which is incorporated into nascent RNA transcripts and can be visualized by conjugation with Alexa-594 via click chemistry (Jao and Salic, 2008). Since ~90% of the total RNA in human cells corresponds to rRNA (Palazzo and Lee, 2015), and we observed that the bulk of the 5-EU-containing RNA is localized in the nucleoli of siControl-transfected cells (Fig. 3C,E), quantification of 5-EU intensity was performed in whole nuclei. To quantify 5-EU intensity, confocal images were segmented on the DAPI channel using CellProfiler (Stirling et al., 2021) to detect individual nuclei, and several measures of signal intensity (e.g. maximum intensity of 5-EU signal, standard deviation of the 5-EU signal) were obtained per nucleus. Given that the nucleolus is the region of the nucleus that shows the highest level of 5-EU incorporation, we reasoned that comparing the maximum intensity of 5-EU signal per nuclei reflects the levels of maximum rDNA transcription in the cells. Comparisons using this metric showed a significant decrease in maximum 5-EU intensity in cells transfected with siZNF16-E compared to siControl (Fig. 3D,F). Another way to quantify the extent of rDNA transcription is by looking at the difference between the high transcription regions (nucleoli) and the low transcription regions (nucleoplasm) by measuring the standard deviation of the 5-EU intensities in the nucleus. This analysis showed a significant decrease in the standard deviation of the intensity in siZNF16-E versus siControl-transfected cells. A lower standard deviation indicates a more homogenous 5-EU staining, as observed in siZNF16-E-treated cells that have lower 5-EU staining in the nucleoli that is closer in value to the staining in the nucleoplasm (Fig. 3C,E). These metrics were chosen because they capture more accurately the changes in 5-EU incorporation in the nucleoli as compared to the measures that quantify 5-EU incorporation in the whole nucleus such as the mean intensity, which also showed a significant decrease after siZNF16-E transfection (data not shown).

In addition, segmentation of the nucleoli was attempted on the 5-EU channel using CellProfiler in order to compare the number, size and intensity of the nucleoli. However, due to the large difference in 5-EU intensity levels between siControl and siZNF16-E cells, we were unable to find segmentation settings that worked well to identify all nucleoli in both treatments. Quantification of the number of nucleoli per nucleus from an attempt using settings that successfully identified all nucleoli in the siControl cells is shown in Fig. 3D,F (right-most graphs). These results show a significant decrease in the number of nucleoli per nucleus in cells treated with siZNF16-E compared to siControl. However, this is likely due to the failure to identify many nucleoli in the siZNF16-E-treated cells due to the low intensity of 5-EU in these cells. Although these results might not be an actual reflection of the number of nucleoli in these cells, they are consistent with a significant decrease in rDNA transcription that results in the inability to segment many nucleoli in siZNF16-E-treated cells. Taken together, these results indicate a role for ZNF16 in promoting rDNA transcription.

## ZNF16 binds preferentially in the intergenic spacer region of the rDNA

Since our previous results indicated that ZNF16 regulates rDNA transcription and given that ZNF16 contains multiple zinc finger

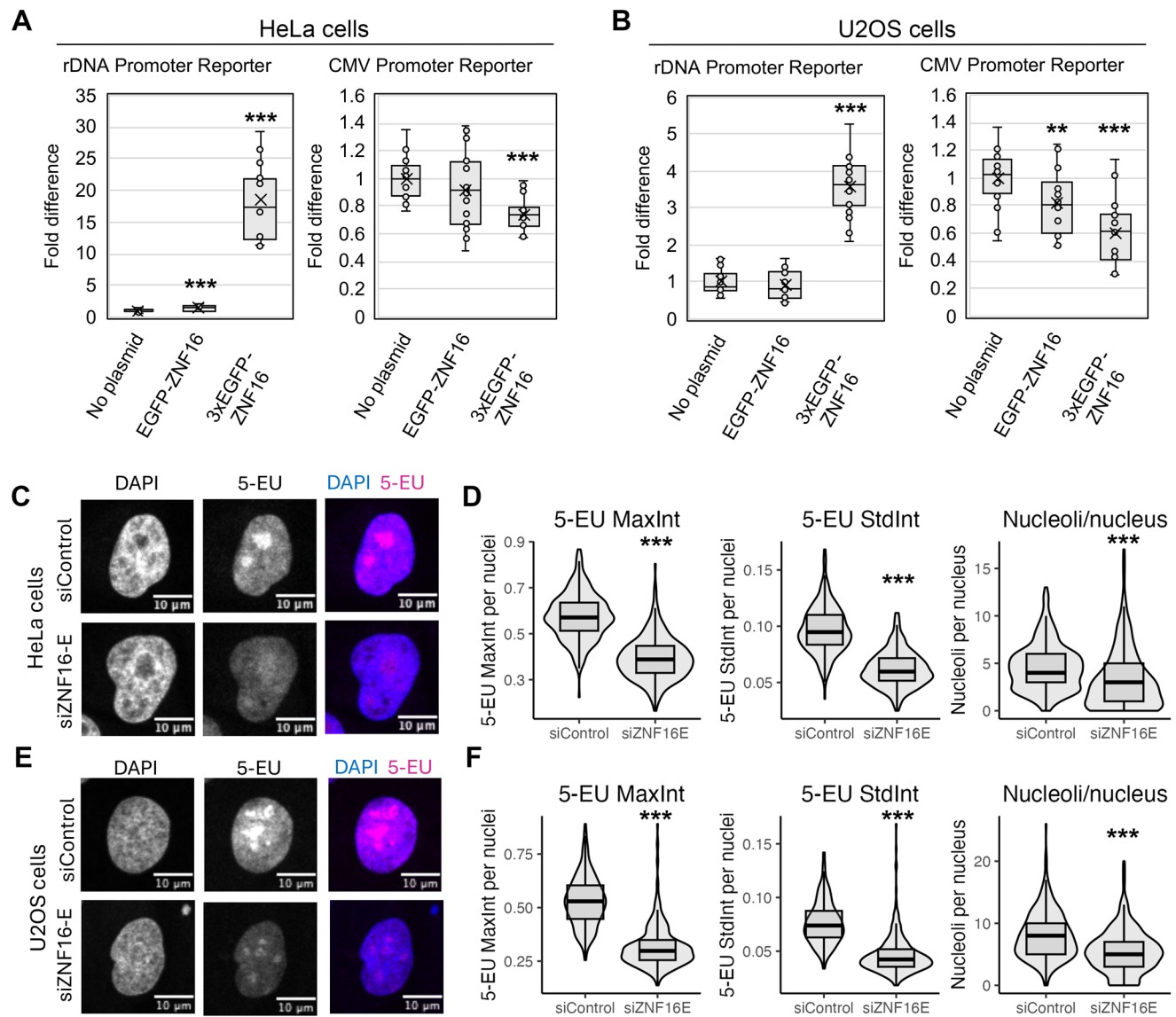

**Fig. 3. ZNF16 regulates rDNA transcription.** (A,B) Luciferase assays after co-transfection of HeLa (A) or U2OS (B) cells with plasmids containing EGFP, EGFP-ZNF16 or 3xEGFP-ZNF16 in combination with either an rDNA transcription luciferase reporter or a CMV promoter luciferase reporter. Results from three independent experiments with at least five technical repeats per experiment. Comparisons to the no plasmid sample performed by *t*-test. (C-F) 5-EU incorporation assay for HeLa (C,D) or U2OS cells (E,F). (C,E) Representative images of HeLa (C) or U2OS cells (E) visualized with Alexa594-conjugated 5-EU and counterstained with DAPI. (D,F) Quantification of the number of nucleoli per nucleus, maximum intensity of 5-EU per nucleus (5-EU MaxInt) and standard deviation of the 5-EU intensity per nucleus (5-EU StdInt) for HeLa cells (D; siControl *n*=288, siZNF16E *n*=300) or U2OS cells (F; siControl *n*=311, siZNF16E *n*=345). Samples compared using Welch two sample *t*-test. Asterisks denote *P*-values as follows: \*\**P*<0.01, \*\*\**P*<0.001.

motifs, which are commonly associated with DNA binding, we tested whether ZNF16 interacts with the rDNA region by chromatin immunoprecipitation (ChIP). HeLa cells stably expressing 3xEGFP-ZNF16 were crosslinked and incubated with anti-EGFP antibody or control IgG. Binding to different regions of the rDNA was quantified by qPCR using primers targeted to different rDNA regions (Zhu et al., 2010) (Fig. 4A). All regions of the rDNA showed at least 5-fold binding enrichment in anti-GFP ChIPs compared to control IgG (Fig. 4B). Interestingly, primers H27, H36 and H42, which target the second half of the intergenic spacer region (IGS), have greater than 15-fold enrichment compared to IgG (Fig. 4B), indicating that ZNF16 binds preferentially to this region of the IGS. These results indicate that ZNF16 regulates rDNA transcription via binding to the IGS region.

## ZNF16 regulates expression of genes in a diversity of pathways and biological processes

To further explore the function of ZNF16, changes in gene expression after ZNF16 depletion were tested by RNA sequencing (RNA-seq). The siRNA siZNF16-E was selected for this experiment due to its consistent reduction of *ZNF16* mRNA (Fig. 1B) and protein levels (Fig. 1C). Comparison between the non-targeting siControl and siZNF16-E transfected samples identified 2833 differentially expressed genes out of a total 21,291 genes with measured expression (Fig. 5A).

Pathway analysis using iPathwayGuide identified focal adhesion (FA), cytokine-cytokine receptor interaction (CCRI), extracellular matrix-receptor interaction (ECMRI), human papillomavirus infection (HPVI), and pathways in cancer (PC) as the top five pathways affected by ZNF16 depletion (Fig. 5B and Table 1). Many

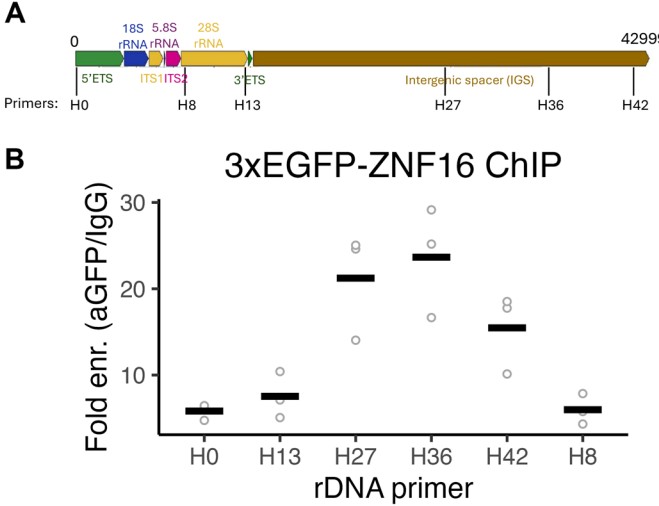

**Fig. 4. ZNF16 binds to the intergenic spacer (IGS) region of the rDNA.** (A) Schematic of the rDNA unit showing the location of the qPCR primers (H0 to H42). ETS, external transcribed spacer; ITS, internal transcribed spacer. (B) Quantification of fold enrichment for different regions in the rDNA unit via chromatin immunoprecipitation (ChIP) with anti-GFP and control IgG antibodies. Individual datapoints (circles) and mean (line) are shown. Data from three independent ChIPs.

of the top ten differentially expressed genes in each pathway (e.g. *BIRC3*, *WNT5B*, *ITGB8*, *NRAS*) are present in more than one pathway, as can be seen in the chord diagram (Fig. 5C).

Gene ontology (GO) analysis was performed using iPathwayGuide with high-specificity and smallest common denominator pruning to identify biological processes and molecular functions associated with ZNF16. The top five biological processes identified with these two methods are shown in Table 2, section A, and the top five molecular functions identified are shown in Table 2, section B. The top biological process and molecular function are both related to extracellular matrix biology, indicating a potential novel function for ZNF16 that remains to be explored.

## ZNF16 depletion affects expression of genes involved in cancer-related pathways

Given that ZNF16 expression and mutations have been associated with gallbladder carcinoma (Ahn et al., 2020) and tongue squamous cell carcinoma (Zhang et al., 2020), we next explored differentially expressed genes that are related to cancer processes. The top twenty differentially expressed genes in pathways in cancer (PC) include the well-known oncogene *NRAS* (Hobbs et al., 2016), the inhibitor of apoptosis *BIRC3*/*cIAP2* (Frazzi, 2021), and *WNT7B*, a component of the WNT/β-catenin pathway (Arensman et al., 2014) (Fig. 6A). The results from the RNA-seq experiment for these three genes were confirmed by qPCR (Fig. 6B), showing statistically significant upregulation of *BIRC3* and downregulation of *NRAS* and *WNT7B* in U2OS cells transfected with siZNF16-E. Another cancer-associated gene that was differentially expressed in the RNA-seq experiment but was not listed in the pathways in cancer list is *EGFR*. Upregulation of *EGFR* after ZNF16 depletion was similarly confirmed by qPCR (Fig. 6C).

Next, we explored the differentially expressed genes included in the CCRI pathway (Fig. 6D). Cytokines are central regulators of immunity and inflammation and play a key role in anti-tumor immunity (Kureshi and Dougan, 2025). The results from the RNA-seq experiment for three genes from the CCRI pathway

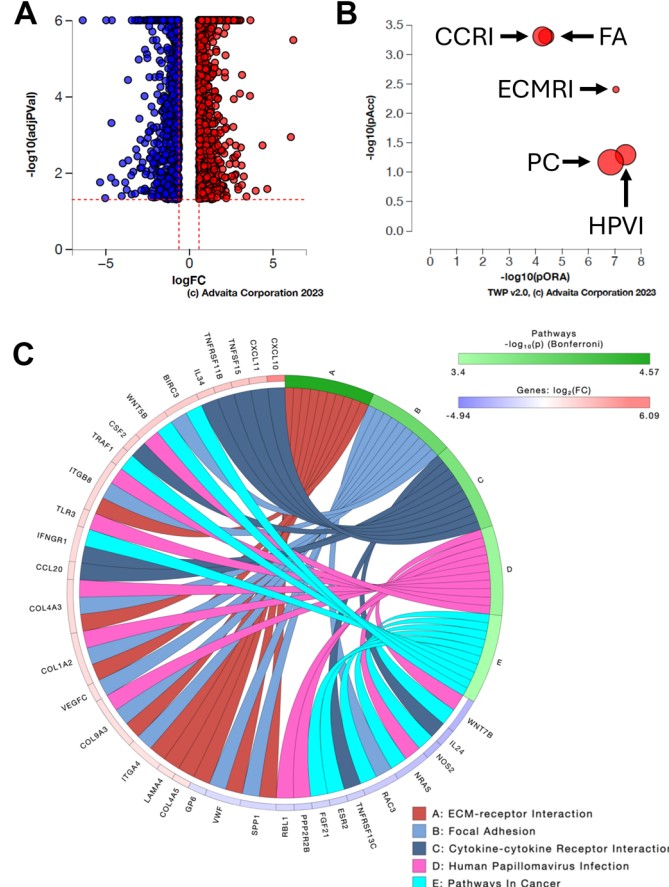

**Fig. 5. Top five pathways affected by ZNF16 depletion.** U2OS cells were transfected with siControl or siZNF16-E siRNAs, and gene expression was measured by RNA-seq. (A) Volcano plot showing the 2883 significantly differentially expressed genes represented by their expression change (logFC) versus the significance of the change [-log10(adjPVal)]. Upregulated genes are shown in red and downregulated genes in blue. (B) Graph showing the top five pathways associated with ZNF16 depletion based on pathway overrepresentation (pORA) and total pathway accumulation (pAcc). Each pathway is represented by a single bubble and the size of the bubble is proportional to the size of the pathway it represents. FA=focal adhesion, CCRI=cytokine-cytokine receptor interaction, ECM=extracellular matrix-receptor interaction, HPVI=human papillomavirus infection, PC=pathways in cancer. (C) Chord diagram of the top ten differentially expressed genes in the five pathways. Each pathway is represented by a different chord color as shown in the legend. Changes in gene expression (log2FC) are depicted in the colors shown in the key.

were confirmed by qPCR (Fig. 6E), showing statistically significant upregulation of *CSF2*, *TNFRSF11B*, and *TNFRSF21* in U2OS cells transfected with siZNF16-E. Another gene involved in immune regulation and associated with poor prognosis in tongue squamous cell carcinoma is *DHRS9* (Shimomura et al., 2018). *DHRS9* was significantly increased in the RNA-seq experiment but was not listed in the CCRI pathway. Upregulation of *DHRS9* after ZNF16 depletion was similarly confirmed by qPCR (Fig. 6F).

Together, our results expand our understanding of ZNF16 function by characterizing ZNF16 as a nucleolar protein that regulates rDNA expression via interaction with the intergenic spacer region of the rDNA unit and providing evidence for other potential roles for ZNF16 in regulating pathways involved in cancer, immune regulation, and extracellular matrix function.

**Table 1. Top five pathways identified by pathway analysis of RNA-seq data after depletion of ZNF16**

| Pathway name | KEGG pathway ID | P-value | P-value (FDR) | P-value (Bonferroni) |
|---|---|---|---|---|
| ECM-receptor interaction (ECMRI) | 04512 | 8.131e-8 | 2.699e-5 | 2.699e-5 |
| Focal adhesion (FA) | 04510 | 3.583e-7 | 5.443e-5 | 1.190e-4 |
| Cytokine-cytokine receptor interaction (CCRI) | 04060 | 4.919e-7 | 5.443e-5 | 1.633e-4 |
| Human papillomavirus infection (HPVI) | 05165 | 9.153e-7 | 7.597e-5 | 3.039e-4 |
| Pathways in cancer (PC) | 05200 | 1.206e-6 | 8.009e-5 | 4.005e-4 |

Data obtained using iPathwayGuide. The P-values were computed using only over-representation analysis. FDR, false discovery rate.

## DISCUSSION

ZNF16 was previously studied in the context of blood cell differentiation but it is a ubiquitously expressed protein (Lorenz et al., 2010; Peng et al., 2006). Our research has confirmed that ZNF16 is expressed in different cancer and non-cancer cell lines and identified a novel role for ZNF16 at the nucleolus as a regulator of rDNA expression. These results are consistent with evidence from affinity purification followed by mass spectrometry that showed interaction between ZNF16 and nucleolar proteins including TCOF1 and UBTF (Schmitges et al., 2016). Our results from the 3xEGFP-ZNF16 ChIP indicate that ZNF16 likely regulates rDNA transcription via binding to the IGS region. Importantly, the ribosomal DNA consists of multiple repeats of the rDNA, one after the other. Given that ZNF16 preferentially binds to the 3′ half of the IGS (Fig. 4B), this region is physically located upstream of the next rDNA sequence. However, the end of the IGS is still ∼4000 bp upstream of the 18S rRNA sequence, indicating that most likely ZNF16 regulates rDNA transcription via long-distance chromatin interactions rather than by directly binding to the rDNA promoter.

Several zinc finger proteins are reported to localize at the nucleolus, including LYAR (Izumikawa et al., 2019), PARP-1 (Meder et al., 2005), PHF12 (also known as PF1) (Graveline et al., 2017), ZNF274 (Yano et al., 2000), ZNF692 (Lafita-Navarro et al., 2023) and ZPR1 (Galcheva-Gargova et al., 1998). PARP-1, PHF12, ZPR1, and ZNF692 are involved in rRNA processing and/or ribosomal biogenesis (Galcheva-Gargova et al., 1998; Graveline et al., 2017; Izumikawa et al., 2019; Lafita-Navarro et al., 2023), while LYAR and ZNF274 are transcription regulators (Begnis et al., 2024; Izumikawa et al., 2019). These two rDNA transcriptional regulators have been shown to act in different ways: ZNF274 is a transcriptional repressor that sequesters gene clusters to transcriptionally inactive perinucleolar regions (Begnis et al., 2024), while LYAR directly binds

near the rDNA promoter and promotes histone acetylation via recruitment of the BRD2-KAT7 complex (Izumikawa et al., 2019). Our results indicate that ZNF16 is another rDNA transcriptional regulator that likely works via interaction with the IGS. However, whether the interaction is direct or indirect remains to be determined.

In addition to the role of ZFN16 at the nucleolus, RNA-seq analysis after depletion of ZNF16 indicates that changes in ZNF16 levels impact gene expression of pathways involved in immune regulation and cancer. ZNF16 depletion results in changes in expression of well-characterized oncogenes such as *NRAS* and *EGFR*. Further exploration of this role of ZNF16 will shed light on its association with tumor progression (Ahn et al., 2020; Lee et al., 2021). Lastly, the GO term analysis also shows potential involvement of ZNF16 with extracellular membrane functions, a potential function that remains to be explored.

## MATERIALS AND METHODS
### Cell culture and drug treatments
U2OS (ATCC), HeLa Tet-On, DLD1, RPE1-hTERT, and Hct116 cells (kind gift from H. Yu) were grown in a humidified incubator at 37°C with 5% $CO_2$ in Dulbecco's modified Eagle's medium with 4.5 g/l glucose, L-glutamine, and pyruvate (DMEM; Corning), supplemented with 10% fetal bovine serum (FBS; HyClone). All cell lines were initially tested for *Mycoplasma* contamination via PCR when received and were monitored routinely via Hoechst staining. Drug treatments: Actinomycin-D (40 μM) and DMSO.

### Plasmids and siRNA transfections
Plasmid pIRESpuro-EGFP-ZNF16 (EGFP-ZNF16), which contains a single copy of enhanced Green Fluorescent Protein (EGFP) fused to ZNF16, was generated by PCRing the open reading frame (ORF) and 3′UTR of *ZNF16* from a pBeloBAC11 containing the full *ZNF16* gene (CTD-2012A17; Invitrogen), adding the FseI and AscI sites for subcloning

**Table 2. Top five biological processes and molecular functions identified by gene ontology (GO) analysis**

| A. Top five biological processes | | | | |
|---|---|---|---|---|
| **Pruning type: high specificity** | | | **Pruning type: smallest common denominator** | |
| GO term | P-value | | GO term | P-value |
| Extracellular matrix organization | 4.946e-4 | | Extracellular matrix organization | 1.789e-5 |
| Cellular response to retinoic acid | 0.126 | | Response to retinoic acid | 0.068 |
| Endodermal cell differentiation | 0.126 | | Angiogenesis | 0.176 |
| Regulation of transcription involved in G1/S transition of mitotic cell cycle | 0.176 | | Regulation of transcription involved in G1/S transition of mitotic cell cycle | 0.176 |
| Positive regulation of cell migration | 0.226 | | Ventricular septum morphogenesis | 0.210 |
| **B. Top five molecular functions** | | | | |
| **Pruning type: high specificity** | | | **Pruning type: smallest common denominator** | |
| GO term | P-value | | GO term | P-value |
| Extracellular matrix structural constituent conferring tensile strength | 0.177 | | Extracellular matrix structural constituent | 0.002 |
| Integrin binding | 0.180 | | Integrin binding | 0.180 |
| DNA replication origin binding | 0.327 | | Identical protein binding | 0.196 |
| Extracellular matrix structural constituent | 0.335 | | DNA replication origin binding | 0.196 |
| Chemorepellent activity | 0.335 | | TAP binding | 0.196 |

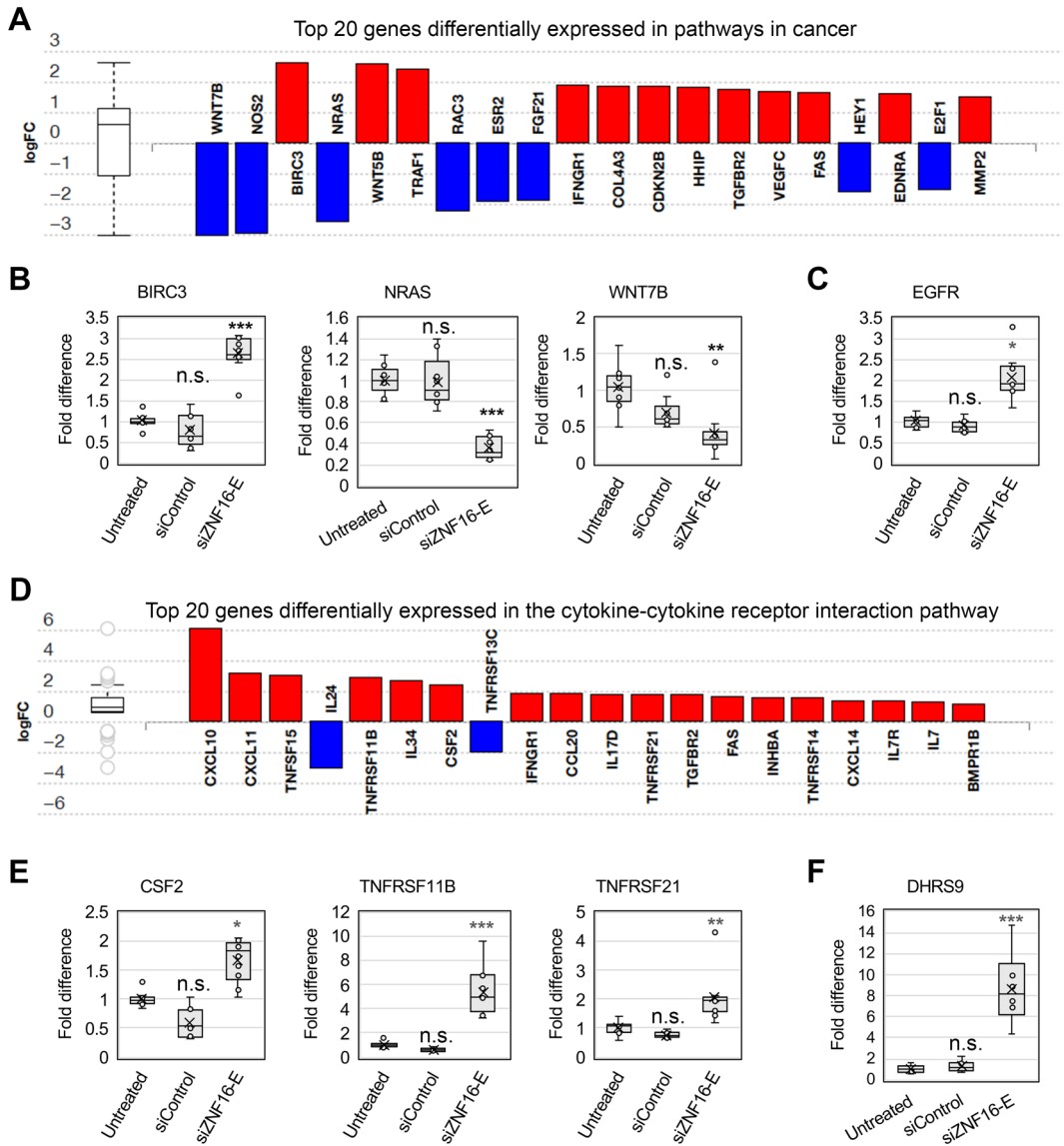

**Fig. 6. ZNF16 depletion affects expression of genes in cancer-associated pathways.** (A) Top 20 genes differentially expressed in pathways in cancer (KEGG: 05200). Upregulated genes shown in red, downregulated genes in blue. (B) Independent confirmation of gene expression changes by qPCR for three genes in the pathway shown in (A). Box plots represent the results from three experiments. Fold difference normalized to the average of the untreated sample. (C) Fold difference in the expression of EGFR. Box plot represents the average of three experiments. Fold difference normalized to the average of the untreated sample. (D) Top 20 genes differentially expressed in the cytokine-cytokine receptor interaction (CCRI) pathway (KEGG: 04060). Upregulated genes shown in red, downregulated genes in blue. (E) Independent confirmation of gene expression changes by qPCR for three genes in the CCRI pathway shown in D. Box plots represent the results from three experiments. Fold difference normalized to the average of the untreated sample. (F) Fold difference in the expression of DHRS9. Box plot represents the average of three experiments. Fold difference normalized to the average of the untreated sample. All comparisons in B, C, E and F were performed by Dunnett's test using the untreated sample as the reference group. Asterisks denote *P*-values as follows: \**P*<0.05, \*\**P*<0.01, \*\*\**P*<0.001. n.s., not significant.

into pIRESpuro-EGFP (EGFP; kind gift from H. Yu). The pIRESpuro-3xEGFP-ZNF16 plasmid (3xEGFP-ZNF16), which contains three copies of EGFP fused to ZNF16, was generated by subcloning of the *ZNF16* gene from the pIRESpuro-EGFP-ZNF16 plasmid using FseI/AscI. Plasmids pHrD-IRES-Luc (human rRNA promoter-luciferase reporter) and pIRES-Luciferase were kind gifts from Ghoshal et al. (2004) and Z. Karamysheva, respectively. Plasmids were transfected at a final concentration of 0.4 ng/µl using Lipofectamine 2000 (Life Technologies). The stably transfected 3xEGFP-ZNF16 cell line was produced by transfection of the pIRESpuro-3xEGFP-ZNF16 plasmid in HeLa Tet-On cells, selection with 0.5 µg/ml puromycin, clone picking and confirmation of gene expression via microscopy.

The sequences of the siRNAs are as follows: siControl (AccuTarget Negative Control siRNA (BioRP SN-1002, VWR 95030-562), siZNF16-E 5′-AAACUAUGCUGGUGAUGUU-3′ (Dharmacon), and siZNF16-3UTR

5′-UGACGUUUGGUUUGAGAUA-3′ (ON-TARGET Plus, Horizon Discovery). siRNAs were transfected at a final concentration of 10 nM using Lipofectamine RNAiMAX (Life Technologies) according to manufacturer recommendations.

### Cell viability assays
Cells were transfected with the indicated siRNAs for 72 h, then incubated with OZBlue (OZ Biosciences) for at least 30 min, and fluorescence was measured using a FLx800 plate reader (BioTek). Background fluorescence from blank wells was subtracted, and fluorescence was normalized to untreated cells.

### Western blotting
Cells were washed in PBS and lysed in lysis buffer (250 mM SDS, 82.5 mM Tris Base, 30% glycerol, 1.5 mM bromophenol blue, and 43.7 mM DTT),

sonicated, and boiled for 10 min. Samples were separated by SDS-PAGE and transferred to a nitrocellulose membrane (GE Healthcare Life Sciences) using a semi-dry transfer apparatus (Bio-Rad). The membranes were incubated with primary antibodies against ZNF16 (rabbit anti-ZNF16 at 1:50; Sigma-Aldrich, HPA061835-100UL) and tubulin (mouse anti-tubulin ascites at 1:100; kindly provided by S. Roychowdhury) dissolved in 5% non-fat dry milk in TBS-T (1× TBS+0.01% Tween 20) overnight at 4°C. Membrane was then incubated with secondary antibodies anti-rabbit IRDye-680RD (LICOR, 926-68070; 1:2000) and goat anti-mouse IgG (H&L) horseradish peroxidase (HRP)-conjugated antibody (ImmunoReagents, GTXMU003DHRPX; 1:10,000) at room temperature for 30 min, followed by incubation with Clarity Western Luminol/Enhancer Reagent (Bio-Rad) for visualization of the HRP-conjugated antibody. The membranes were imaged using a LI-COR Odyssey imager.

### Immunostaining

Cells were cultured in eight-well chamber slides (Nunc Lab-Tek II or Falcon) and treated as described. Cells were fixed with 4% paraformaldehyde (PFA) for 20 min at room temperature. Primary antibodies were diluted in blocking buffer (0.2% Triton X-100 in PBS with 3% bovine serum albumin) and incubated at 4°C overnight. Secondary antibodies were diluted in blocking buffer and incubated at room temperature, in the dark for 30 min. Cells were washed with 0.2% Triton X-100 in PBS, counterstained with DAPI, and mounted using Vectashield anti-fade mounting media (VectorLabs). Primary antibodies used were as follows: rabbit anti-ZNF16 (1:50; Sigma-Aldrich, HPA061835-100UL), rabbit anti-GFP (1:500; Novus), mouse anti-Ki67 (1:250; BD Biosciences, 610968), rabbit anti-UBF (1:400; Novus, NBP1-82545-25ul), and mouse anti-NPM1 (1:800; Protein Tech, 60096-1-Ig). Secondary antibodies used were as follows: Alexa Fluor 488 donkey anti-rabbit IgG, Alexa Fluor 568 donkey anti-mouse IgG, and donkey anti-mouse Alexa Fluor 647 IgG (1:500; Life Technologies).

### 5-EU incorporation assay

Cells were transfected with siRNAs for 72 h, then incubated with 0.5 mM 5-EU for an hour. Next, cells were fixed with either 4% PFA or cold methanol and stained via click reaction with a fluorescent azide using the RNA Synthesis Assay Kit (Abcam, ab228561) or the Click-IT RNA AlexaFluor 594 RNA synthesis imaging kit (Invitrogen), according to the manufacturer's instructions. Cells were then washed overnight with 0.2% Triton X-100 in PBS, counterstained with DAPI, and mounted using Vectashield anti-fade mounting media (VectorLabs).

### Confocal microscopy and image analysis

Samples were imaged with an LSM 700 confocal microscope (Zeiss), equipped with an EC Plan-NEOFLUAR 63×/1.25 N.A. oil immersion objective and ZEN 2009 software (Zeiss), or a TCS SPE-II confocal microscope (Leica) equipped with ACS APO oil immersion objectives (40×/1.15 NA Oil CS; 63×/1.30 NA) and LASX software. At least 12 z-stacks were acquired per field. Images were then semi-automatically processed in ImageJ using a macro for z-stack projection (kind gift from B. Bell). Nuclei and nucleoli were segmented and quantified using CellProfiler (Stirling et al., 2021). The data were analyzed using Excel, R-studio and/or JMP.

### ChIP

HeLa Tet-On and EGFP-ZNF16 HeLa Tet-On cells were crosslinked with 1% formaldehyde, collected, washed, and lysed, then the nuclei were isolated and the chromatin was immunoprecipitated with 5 μg of antibodies against UBF, GFP, or control IgG using a Magnetic ChIP Kit (ThermoScientific Pierce) according to the manufacturer's protocol. After DNA elution from the beads, the immunoprecipitated rDNA was quantified by qPCR using Power Up SYBR Green Mastermix (Applied Biosystems) and previously described primers spanning the rDNA repeat (Zhu et al., 2010).

### qPCR

Total RNA was extracted from cells using TRIzol with the Direct-zol RNA Miniprep Kit (Zymo Research) according to the manufacturer's protocol. RNA extracts were reverse transcribed using the High-Capacity cDNA Kit (Applied Biosystems). cDNA samples were subjected to qPCR in an Applied Biosystems 7900HT qPCR, using either TaqMan probes and the TaqMan Gene Expression Master Mix (ThermoFisher Scientific), or DNA primers and the Luna Universal Probe qPCR Mastermix (M3004, New England Biosciences).

### RNA-seq

Cells were treated with siControl and siZNF16-E siRNAs for 72 h, then lysed in TRIzol, frozen and shipped to the University of Arkansas for Medical Sciences (UAMS) Genomics Core Facility. Triplicate samples per siRNA were analyzed. Quality control for the 76 base pairs single-end raw reads was ensured using Trimmomatic (Bolger et al., 2014) to perform the following steps: (1) remove Illumina adapter and PCR primer sequences, (2) remove leading and trailing bases with low quality, (3) scan reads with a 4-base wide sliding window and cut when the average quality score per base drops below 15, and (4) drop reads shorter than 36 bases long. Reads surviving the quality control criteria were aligned to the human genome model hg19 using Tophat (Trapnell et al., 2012), allowing two mismatches. Alignments were quantified per gene using featureCounts from package Subread (Liao et al., 2014). Genes with zero counts in five or more out of six samples were deemed unexpressed and discarded, leaving 30,245 expressed genes. Initial differential expression (DE) analysis with paired-sample design was performed using Wald test from Bioconductor package DESeq2 (Love et al., 2014). The RNA-seq read summary and the full results from DESeq2 are provided in Table S1. RNA-seq results were further analyzed using iPathwayGuide (AdvaitaBio) as follows: differentially expressed genes were identified as those with an adjusted $P$-value <0.05 and $\log_2$ fold change of >0.6, and the data were analyzed for enrichment of metabolic pathways and diseases using the Kyoto Encyclopedia of Genes and Genomes (KEGG) database (Release 96.0+/11-21 November 2020), gene ontologies from the Gene Ontology Consortium database (14 October 2020), miRNAs from miRbase (MIRBASE Version 22.1, 10/18) and the TARGETSCAN database (human version 7.2), network of regulatory relations from BioGRID (v4.0.189, 25 August 2020), chemical/drugs/toxicants from the Comparative Toxiccogenomics Database (July 2020).

### Acknowledgements

Thanks to Dr Sid Das and all members of the Das and Diaz-Martinez laboratories for support and productive discussions, R. Warrington for assistance with cloning, A. Levario for generation of stable cell line, B. Bell for the ImageJ macro, M. Cannata for assistance with the 5-EU incorporation assay, and M. Blankenfeld and L. LeBlanc for assistance with qPCR.

### Competing interests

The authors declare no competing or financial interests.

### Author contributions

Conceptualization: L.A.D.-M.; Data curation: L.A.D.-M.; Formal analysis: C.L.G., L.A.E.Q., J.P., M.J.A., Y.R., G.V.G., N.S.R., L.A.D.-M.; Funding acquisition: N.S.R., L.A.D.-M.; Investigation: C.L.G., L.A.E.Q., J.P., M.J.A., A.P.L., G.V.G., L.A.D.-M.; Methodology: Y.R., G.V.G.; Project administration: L.A.D.-M.; Resources: A.P.L., Y.R., G.V.G., N.S.R.; Supervision: L.A.D.-M.; Validation: C.L.G., L.A.E.Q., J.P., M.J.A., L.A.D.-M.; Visualization: C.L.G., L.A.E.Q., L.A.D.-M.; Writing – original draft: C.L.G., L.A.E.Q., L.A.D.-M.; Writing – review & editing: L.A.E.Q., M.J.A., Y.R., L.A.D.-M.

### Funding

This work was supported by M.J. Murdock Charitable Trust grant #202016503 to L.A.D.-M. Sequencing at the UAMS Genomics Core facility and bioinformatics support were respectively provided through the Research Technology Core and Bioinformatics Research Support Core of the Arkansas INBRE program, by a grant from the National Institute of General Medical Sciences and by P20 GM103429 from the National Institutes of Health (NIH). Undergraduate student research was partially supported by Gonzaga Science Research Program and the Cell Biology Education Consortium via National Science Foundation (NSF) awards #18270660 and #2316122 to N.S.R. The contents of this article are solely the responsibility of the authors and do not necessarily represent the official views of the NIH or NSF. Open Access funding provided by Gonzaga University. Deposited in PMC for immediate release.

### Data and resource availability

All relevant data and details of resources can be found within the article and its supplementary information.

**Peer review history**

The peer review history is available online at https://journals.biologists.com/bio/lookup/doi/10.1242/bio.062336.reviewer-comments.pdf

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
