## [Peer Review File · Biology Open]

ZNF16 is a nucleolar-associated protein that regulates expression of the rDNA and cancer-associated genes

Chelsea L. George, Laura A. Espinoza Quevedo, Jason Paratore, Matthew J. Alcaraz, Arlene P. Levario, Yasir Rahmatallah, Galina V. Glazko, Nathan S. Reyna and Laura A. Diaz-Martinez

DOI: 10.1242/bio.062336

Editor: Christopher A. Maher

Review timeline

Original submission:	31 October 2025
Editorial decision:	10 November 2025
First revision received:	24 November 2025
Editorial decision:	3 December 2025
Second revision received:	3 December 2025
Accepted:	15 December 2025

Original submission

First decision letter

MS ID#: bio.062336

MS Title: ZNF16 is a nucleolar-associated protein that regulates expression of the rDNA and cancer-associated genes

Authors: Chelsea L. George, Laura A. Espinoza Quevedo, Jason Paratore, Matthew J. Alcaraz, Arlene P. Levario, Yasir Rahmatallah, Galina V. Glazko, Nathan S. Reyna and Laura A. Diaz-Martinez

I have now reached a decision on the above manuscript.

The reviewer reports are shown at the bottom of this email.

As you will see, the reviewers gave favourable reports, but raised some critical points that will require amendments to your manuscript. I hope that you will be able to carry these out, because we would like to be able to accept your paper.

At this stage, we also ask you to ensure your manuscript complies with our formatting guidelines - please see our manuscript preparation guidelines for details. Provided you are able to fully address the referees' comments, we are positive about publication of your paper (we accept over 95% of revision submissions) and therefore hope you won't mind any extra work involved in reformatting your manuscript at this point.

Please upload both a 'clean' version of your Word file, along with a highlighted version clearly showing where you have made changes in the revised manuscript. Please avoid using 'Track changes' in Word files as these are lost in PDF conversion.

I should be grateful if you would also provide a point-by-point response detailing how you have dealt with the points raised by the reviewers in the 'Response to Reviewers' box. Please attend to all of the reviewers' comments. If you do not agree with any of their criticisms or suggestions please explain clearly why this is so.

Reviewer 1

Comments for the author

This article is very strong with some very interesting findings and exciting results. I have minor recommendations to help with the clarity of the data and reasons for using certain plasmids/procedures etc.

1. Hypothesis: Is there a reason there is a reference in the hypothesis? The hypothesis referenced (Peng et al. 2006) has done a lot of work with ZNF16 and has certainly laid some of the ground work for your research and data. But if this is the hypothesis of your lab group, I do not think the immediate reference after your hypothesis to another lab is necessary.
2. Materials and Methods: What is the difference between EGFP-ZNF16 and 3xEGFP-ZNF16? I do not see any description that defines the difference between these 2. They also behave differently in some results sections and I am unclear what the difference is between these and why the results are different.
3. Results Section (ZNF16 is expressed and promotes cell viability in a variety of cell lines): The second paragraph discusses the findings in Figure 1B and states siZNF16-E most consistently reduced ZNF16 siRNAs compared to the other two siRNAs. Do you mean to say reduced ZNF16 RNA expression? This phrasing is unclear. The third paragraph discusses the findings I believe of Figure 1D although it is not listed. Please clarify.
4. Results Section (ZNF is enriched at the nucleoli): The first paragraph discusses EGFP-ZNF16 and 3xEGFP-ZNF16? What is the difference between these 2 and why are they behaving differently? Please clarify what freckled localization is and where you are seeing it in images (bottom 2 rows of Figure 2B). The fourth paragraph discusses the findings I believe of Figure 2G although it is not listed. Please clarify.
5. Results Section (ZNF16 regulates rDNA expression): Expression of 3xEGFP-ZNF16 which was shown to be enriched at the nucleolus (Fig 2A-B). Is this supposed to be Figure 3A-B?
6. Figure 2: When describing the findings of Figure 2 in the results section, please clarify which boxes within Figure 2A, B, D, and G you are referring to. There are lots of boxes and sometimes it is unclear what I am supposed to look at to verify your written results match your visual data.

Reviewer 2

Comments for the author

The authors present an investigation of ZNF16 localization and function in cancer cells. Overall the data presented nicely illustrate that ZNF16 is highly expressed in several cancer cell lines and that ZNF16 is localized to the nucleolus using several different methods. Functionally, they show that ZNF16 can regulate rDNA transcription through binding to the intergenic spacer region. RNAseq analysis identifies several pathways that are differentially regulated in response to ZNF16 knockdown. Overall the manuscript is well written and most of the conclusions are supported by the data. There are a few points that should be addressed by the authors.

1. The authors do not include the statistical methods used for data analysis.
2. Lines 380-381 should include the cell line that was being used.
3. For the RNAseq analysis a log2fold change of 0.6 was used as a cut-off. Is there a strong rationale for not using a higher cut-off such as 1.0, which is more standard.
4. NRas is described as a tumor suppressor, but is a well known oncogene. This should be corrected within the results and discussion.
5. Within the Discussion (lines 437-439), a manuscript by Schmitteges et al 2016 is referenced and refers to co-IPs from that study. This data is not readily obtained by looking at the manuscript. Please clarify that this statement is correct.
6. While the validation of the RNAseq data by qPCR is a good control, it would be more significant to show that these genes are regulated by siRNA KD of ZNF16 in a separate cell line.

Reviewer's Responses to Questions

Experimental quality

Does each figure have the proper controls?

If 'No', please indicate reasons in Comments for Author box below.

Reviewer #1:

- Yes

Reviewer #2:

- Yes
-

Were the data analyzed using appropriate statistical tests?

If 'No', please indicate reasons in Comments for Author box below.

Reviewer #1:

- Yes

Reviewer #2:

- No
-

Reproducibility

Were experiments performed using adequate number of biological replicates?

If 'No', please indicate reasons in Comments for Author box below.

Reviewer #1:

- Yes

Reviewer #2:

- Yes
-

Does the methods section provide sufficient detail to permit reproducibility?

If 'No', please indicate reasons in Comments for Author box below.

Reviewer #1:

- Yes

Reviewer #2:

- Yes
-

Completeness

Are the manuscript's conclusions supported by the data?

If 'No', please indicate reasons in Comments for Author box below.

Reviewer #1:

- Yes

Reviewer #2:

- Yes
-

Scholarship

Do the authors cite and discuss the merits of data that would argue for and against their conclusion?

If 'No', please indicate reasons in Comments for Author box below.

Reviewer #1:

- Yes

Reviewer #2:

- Yes
-

Does the manuscript title & abstract accurately reflect the contents of the manuscript, without hyperbole?

If 'No', please indicate reasons in Comments for Author box below.

Reviewer #1:

- Yes

Reviewer #2:

- Yes

First revision

Author response to reviewers' comments

Dear Editor and Reviewers,

Comments from the Reviewers:

Reviewer 1: *This article is very strong with some very interesting findings and exciting results. I have minor recommendations to help with the clarity of the data and reasons for using certain plasmids/procedures etc.*

1. *Hypothesis: Is there a reason there is a reference in the hypothesis? The hypothesis referenced (Peng et al. 2006) has done a lot of work with ZNF16 and has certainly laid some of the ground work for your research and data. But if this is the hypothesis of your lab group, I do not think the immediate reference after your hypothesis to another lab is necessary.*

The authors agree with the reviewer. The reference has been removed (line 110).

2. *Materials and Methods: What is the difference between EGFP-ZNF16 and 3xEGFP-ZNF16? I do not see any description that defines the difference between these 2. They also behave differently in some results sections and I am unclear what the difference is between these and why the results are different.*

Thank you for bringing up this very important point, which we failed to address in the original manuscript. The difference is that the EGFP-ZNF16 plasmid has one copy of EGFP followed by ZNF16, while the 3xEGFP-ZNF16 plasmid has three copies of EGFP, one after the other, followed by ZNF16. This has been stated in the methods section (lines 368-375) as well as the results section (lines 144-147). We believe the difference in the size of the tag and/or the folding properties of one vs three EGFPs might account for the difference in localization of the ZNF16 fusion proteins. This has been stated in lines 153-155. By comparing to endogenous ZNF16, which also shows nucleolar enrichment, we concluded that the pattern of localization for 3xEGFP-ZNF16 more accurately mirrors the endogenous protein. We now emphasize this in lines 171-173.

3. *Results Section (ZNF16 is expressed and promotes cell viability in a variety of cell lines): The second paragraph discusses the findings in Figure 1B and states siZNF16-E most consistently reduced ZNF16 siRNAs compared to the other two siRNAs. Do you mean to say reduced ZNF16 RNA expression? This phrasing is unclear. The third paragraph discusses the findings I believe of Figure 1D although it is not listed. Please clarify.*

Thank you for pointing out these mistakes. Indeed, as the reviewer indicates, the text describing the results from Figure 1B should read “siZNF16-E most consistently reduced ZNF16 mRNA”, not siRNAs. The sentence has been corrected (line 131). Similarly, Figure 1D has been listed in the third paragraph as suggested by the reviewer (lines 134-141).

4. *Results Section (ZNF is enriched at the nucleoli): The first paragraph discusses EGFP-ZNF16 and 3xEGFP-ZNF16? What is the difference between these 2 and why are they behaving differently? Please clarify what freckled localization is and where you are seeing it in images (bottom 2 rows of Figure 2B). The fourth paragraph discusses the findings I believe of Figure 2G although it is not listed. Please clarify.*

We agree with the reviewer that the difference between EGFP-ZNF16 and 3xEGFP-ZNF16 was not clearly described. As mentioned in response to point #1, we have added explanations of the difference between these two plasmids in the methods section (lines 368-375) as well as the results section (lines 144-147). We also have now provided further description of the results in lines 143-173.

In addition, a reference to Figure 2G was added on line 186.

5. *Results Section (ZNF16 regulates rDNA expression): Expression of 3xEGFP-ZNF16 which was shown to be enriched at the nucleolus (Fig 2A-B). Is this supposed to be Figure 3A-B?*

The sentence the reviewer is referring to (lines 200-202): “Expression of 3xEGFP-ZNF16, which was shown to be enriched at the nucleolus (Fig. 2A-B), significantly increased luciferase activity in both U2OS and HeLa cells (Fig. 3A-B).” has two parts. The first part is referring back to the results shown in Figure 2A-B, which show 3xEGFP-ZNF16 is enriched in the nucleolus, as a reminder to the reader of the difference in localization between EGFP-ZNF16 and 3xEGFP-ZNF16. The second part refers, as the reviewer points out, to the results from Figure 3A-B, which show increased luciferase activity for the rDNA-promoter-driven luciferase reporter. To clarify this further, we have added references to Figure 2 earlier in the paragraph (lines 193-196), to emphasize that we are first revisiting the results of ZNF16 localization, which we believe is relevant and directly related to the difference in luciferase expression shown in Figure 3.

6. *Figure 2: When describing the findings of Figure 2 in the results section, please clarify which boxes within Figure 2A, B, D, and G you are referring to. There are lots of boxes and sometimes it is unclear what I am supposed to look at to verify your written results match your visual data.*

Thank you for this suggestion. We have modified this paragraph to provide additional guidance to the reader in all the figure references within this paragraph (lines 143-163).

Reviewer 2: *The authors present an investigation of ZNF16 localization and function in cancer cells. Overall the data presented nicely illustrate that ZNF16 is highly expressed in several cancer cell lines and that ZNF16 is localized to the nucleolus using several different methods.*

Functionally, they show that ZNF16 can regulate rDNA transcription through binding to the intergenic spacer region. RNAseq analysis identifies several pathways that are differentially regulated in response to ZNF16 knockdown. Overall the manuscript is well written and most of the conclusions are supported by the data. There are a few points that should be addressed by the authors.

1. *The authors do not include the statistical methods used for data analysis.*

Thank you for this observation. We agree that this is important information that we had not provided in the previous manuscript. We have now listed the statistical methods used for data analysis within every figure legend to ensure this information is provided for each experiment.

2. *Lines 380-381 should include the cell line that was being used.*

We have added the word HeLa at the beginning of the sentence (line 256, please note that the line numbers changed because the Materials and Methods section was moved to the back of the manuscript).

3. *For the RNAseq analysis a log2fold change of 0.6 was used as a cut-off. Is there a strong rationale for not using a higher cut-off such as 1.0, which is more standard.*

The reviewer brings up an important point: the choice of log₂ fold change (log₂FC) threshold in RNAseq analysis has been long debated. A study done in yeast showed that when using triplicate samples, a log₂FC threshold of 0.5 resulted in a true positive rate (TPR) of 0.65, while a log₂FC threshold of 1 resulted in a TPR of 0.8 (Schurch et al., 2016). These results are consistent with the idea that higher thresholds result in higher TPR. However, the use of high thresholds also has drawbacks, as stated by Chen et al., “the most serious drawback is that the most differentially expressed genes (DEGs), meaning those with the lowest *P*-values or highest log-fold changes, may not necessarily be most relevant or explanatory for the phenotypes or conditions of interest in a study.” (Chen et al., 2023). Biologically speaking, a larger fold change does not necessarily imply a larger effect on a given process. Thus, we reasoned that by combining a lower log₂FC threshold with pathway enrichment analysis would allow us to on one hand cast a wider net by identifying more differentially expressed genes (which has the risk of lowering the TPR), while also allowing us to focus only on those genes that belong to pathways with other differentially expressed genes (which increases the TPR).

4. *NRas is described as a tumor suppressor, but is a well known oncogene. This should be corrected within the results and discussion.*

Thank you for pointing out this mistake. This was corrected in the results (line 292) and discussion (line 351).

5. *Within the Discussion (lines 437-439), a manuscript by Schmitges et al 2016 is referenced and refers to co-IPs from that study. This data is not readily obtained by looking at the manuscript. Please clarify that this statement is correct.*

This statement is correct. Schmitges and co-workers reported the results of immunoprecipitation with 118 zinc finger containing proteins, including ZNF16 (Schmitges et al., 2016). Unfortunately, most of the data for the IP was presented in the supplementary materials (specifically, in Supplemental Figure S3 and Supplemental Table S5).

6. *While the validation of the RNAseq data by qPCR is a good control, it would be more significant to show that these genes are regulated by siRNA KD of ZNF16 in a separate cell line.*

Although we do agree with the reviewer that validation of the RNAseq in a separate cell line would be valuable, we think this would be a separate experiment that is not a direct validation of the RNAseq data. The goal of the qPCR experiment the reviewer refers to was to validate the results of the RNAseq experiments using a different method (qPCR) to provide evidence that the RNAseq results were reliable and reproducible by different methods. We believe validation in other cell lines, would be an interesting future experiment that extends beyond the focus of this paper.

In addition to the changes made in response to the reviewers' helpful comments, the following changes were based on the manuscript preparation guidelines provided:

- A Summary Statement was added (lines 43-45).
- Table 2 and 3 were merged into a single Table 2, to comply with the eight display items maximum requirement. The article now contains six figures and two tables. References to Table 2 and Table 3 within the manuscript have been updated to Table2 section A and Table2 section B, respectively.
- The Materials and Methods section was moved back after the Discussion section. Note that this caused all line numbers to change significantly compared to the previously submitted version of the manuscript.
- A statement on Mycoplasma testing for cell lines was added to the Materials and Methods section.
- The acknowledgements section was divided into acknowledgements and funding.
- A 'Data and resource availability' statement was added.
- Catalog numbers for antibodies were added, when available.

Second decision letter

MS ID#: bio.062336R1

MS Title: ZNF16 is a nucleolar-associated protein that regulates expression of the rDNA and cancer-associated genes

Authors: Chelsea L. George, Laura A. Espinoza Quevedo, Jason Paratore, Matthew J. Alcaraz, Arlene P. Levario, Yasir Rahmatallah, Galina V. Glazko, Nathan S. Reyna and Laura A. Diaz-Martinez

I have now reached a decision on the above manuscript.

The reviewer reports are shown at the bottom of this email.

As you will see, the reviewers gave favourable reports, but raised some critical points that will require amendments to your manuscript. I hope that you will be able to carry these out, because we would like to be able to accept your paper.

At this stage, we also ask you to ensure your manuscript complies with our formatting guidelines - please see our manuscript preparation guidelines for details. Provided you are able to fully address the referees' comments, we are positive about publication of your paper (we accept over 95% of revision submissions) and therefore hope you won't mind any extra work involved in reformatting your manuscript at this point.

Please upload both a 'clean' version of your Word file, along with a highlighted version clearly showing where you have made changes in the revised manuscript. Please avoid using 'Track changes' in Word files as these are lost in PDF conversion.

I should be grateful if you would also provide a point-by-point response detailing how you have dealt with the points raised by the reviewers in the 'Response to Reviewers' box. Please attend to all of the reviewers' comments. If you do not agree with any of their criticisms or suggestions please explain clearly why this is so.

Reviewer 1

Comments for the author

All comments have been corrected and clarifications have been input. I am happy with the changes and adjustments that have been made. I am recommending that this article be accepted for publication. Thank you.

Reviewer 2

Comments for the author

The authors have adequately addressed most of the reviewer comments, but from this reviewer's perspective there is still one point that needs further attention.

Reviewer 2, #5.

Within the Discussion (lines 437-439), a manuscript by Schmitges et al 2016 is referenced and refers to co-IPs from that study. This data is not readily obtained by looking at the manuscript. Please clarify that this statement is correct.

Response: This statement is correct. Schmitges and co-workers reported the results of immunoprecipitation with 118 zinc finger containing proteins, including ZNF16 (Schmitges et al.,

2016). Unfortunately, most of the data for the IP was presented in the supplementary materials (specifically, in Supplemental Figure S3 and Supplemental Table S5).

The text reads, "These results are consistent with evidence from co-IPs that showed interaction between ZNF16 and several nucleolar proteins including TCOF1, UBTF, NOL9, and PELP1 (Schmitges et al., 2016)."

First, reading through the referenced manuscript it states that they performed affinity purification mass spectrometry (AP-MS) to identify interacting proteins, not co-IP. Second, looking through Supplemental Table S5, there are further concerns about the interpretation of the data. While TCOF1, UBTF, NOL9, and PELP1 are found in the AP-MS data, the probability score for the NOL9 and PELP1 interaction is very low. Putting these interactions with low probability score in the discussion could lead to others citing this association and perpetuate spreading unverified information as fact.

Reviewer's Responses to Questions

Experimental quality

Does each figure have the proper controls?

If 'No', please indicate reasons in Comments for Author box below.

Reviewer #1:

- Yes

Reviewer #2:

- Yes

Were the data analyzed using appropriate statistical tests?

If 'No', please indicate reasons in Comments for Author box below.

Reviewer #1:

- Yes

Reviewer #2:

- Yes

-

Reproducibility

Were experiments performed using adequate number of biological replicates?

If 'No', please indicate reasons in Comments for Author box below.

Reviewer #1:

- Yes

Reviewer #2:

- Yes

Does the methods section provide sufficient detail to permit reproducibility?

If 'No', please indicate reasons in Comments for Author box below.

Reviewer #1:

- Yes

Reviewer #2:

- Yes

Completeness

Are the manuscript's conclusions supported by the data?

If 'No', please indicate reasons in Comments for Author box below.

Reviewer #1:

-

Reviewer #2:

- Yes

Scholarship

Do the authors cite and discuss the merits of data that would argue for and against their conclusion?

If 'No', please indicate reasons in Comments for Author box below.

Reviewer #1:

- Yes

Reviewer #2:

- Yes

Does the manuscript title & abstract accurately reflect the contents of the manuscript, without hyperbole?

If 'No', please indicate reasons in Comments for Author box below.

Reviewer #1:

- Yes

Reviewer #2:

- Yes

Second revisionAuthor response to reviewers' comments

Dear Editor and Reviewers,

Thank you for the detailed and constructive comments to improve our manuscript. Here is our response to the comment from Reviewer 2:

Reviewer 2: *The authors have adequately addressed most of the reviewer comments, but from this reviewer's perspective there is still one point that needs further attention. Reviewer 2, #5.*

Within the Discussion (lines 437-439), a manuscript by Schmitges et al 2016 is referenced and refers to co-IPs from that study. This data is not readily obtained by looking at the manuscript. Please clarify that this statement is correct.

[Previous authors'] Response: This statement is correct. Schmitges and co-workers reported the results of immunoprecipitation with 118 zinc finger containing proteins, including ZNF16 (Schmitges et al., 2016). Unfortunately, most of the data for the IP was presented in the supplementary materials (specifically, in Supplemental Figure S3 and Supplemental Table S5).

The text reads, "These results are consistent with evidence from co-IPs that showed interaction between ZNF16 and several nucleolar proteins including TCOF1, UBTF, NOL9, and PELP1 (Schmitges et al., 2016)."

First, reading through the referenced manuscript it states that they performed affinity purification mass spectrometry (AP-MS) to identify interacting proteins, not co-IP. Second, looking through Supplemental Table S5, there are further concerns about the interpretation of the data. While TCOF1, UBTF, NOL9, and PELP1 are found in the AP-MS data, the probability score for the NOL9 and PELP1 interaction is very low. Putting these interactions with low probability score in the discussion could lead to others citing this association and perpetuate spreading unverified information as fact.

The authors agree with the reviewer. We thank the reviewer for pointing out the probability score, which we did not take into account in our previous version of the manuscript. The sentence in lines

324-326 has been amended as follows:

“These results are consistent with evidence from affinity purification followed by mass spectrometry (AP-MS) that showed interaction between ZNF16 and nucleolar proteins including TCOF1 and UBTF (Schmitges et al., 2016).”

Third decision letter

MS ID#: bio.062336R2

MS Title: ZNF16 is a nucleolar-associated protein that regulates expression of the rDNA and cancer-associated genes

Authors: Chelsea L. George, Laura A. Espinoza Quevedo, Jason Paratore, Matthew J. Alcaraz, Arlene P. Levario, Yasir Rahmatallah, Galina V. Glazko, Nathan S. Reyna and Laura A. Diaz-Martinez

I am happy to tell you that your manuscript has been accepted for publication in Biology Open, pending our standard publication integrity checks. It was accepted on 15th December 2025.